# Event Linking with Sentential Features from Convolutional Neural Networks

## Abstract

Coreference resolution for event mentions enables extraction systems to process document-level information. Current systems in this area base their decisions on rich semantic features from various knowledge bases, thus restricting them to domains where such external sources are available. We propose a model for this task which does not rely on such features but instead utilizes sentential features coming from convolutional neural networks. Two such networks first process coreference candidates and their respective context, thereby generating latent-feature representations which are tuned towards event aspects relevant for a linking decision. These representations are augmented with lexical-level and pairwise features, and serve as input to a trainable similarity function producing a coreference score. Our model achieves state-of-the-art performance on two datasets, one of which is publicly available. An error analysis points out directions for further research.

## 1 Introduction

*Event extraction* aims at detecting mentions of real-world events and their arguments in text documents of different domains, e.g., news articles. The subsequent task of *event linking* is concerned with resolving coreferences between recognized event mentions in a document, and is the focus of this paper.

Several studies investigate event linking and related problems such as relation mentions spanning multiple sentences. Swampillai and Stevenson (2010) find that 28.5 % of binary relation mentions in the MUC 6 dataset are affected, as are 9.4 % of relation mentions in the ACE corpus from 2003. Ji and Grishman (2011) estimate that 15 % of slot fills in the training data for the "TAC 2010 KBP Slot Filling" task require cross-sentential inference. To confirm these numbers, we analyzed the event annotation of the ACE 2005 corpus and found that approximately 23 % of the event mentions are incomplete on the argument level, with respect to the information in other mentions of the same event instance in the respective document. These numbers suggest that event linking is an important task.

Previous approaches for modeling event mentions in context of coreference resolution (Bejan and Harabagiu, 2010; Sangeetha and Arock, 2012; Liu et al., 2014) make either use of external feature sources with limited cross-domain availability like WordNet (Fellbaum, 1998) and FrameNet (Baker et al., 1998), or show low performance. At the same time, recent literature proposes a new kind of feature class for modeling events (and relations) in order to detect mentions and extract their arguments, i.e., *sentential features* from event-/relation-mention representations that have been created by taking the full extent and surrounding sentence of a mention into account (Zeng et al., 2014; Nguyen and Grishman, 2015; Chen et al., 2015; dos Santos et al., 2015; Zeng et al., 2015). Their promising results motivate our work. We propose to use such features for event coreference resolution, hoping to thereby remove the need for extensive external semantic features while preserving the current state-of-the-art performance level.

Our contributions in this paper are as follows: We design a system for event linking which in a first step models intra-sentential event mentions via the use of convolutional neural networks for the integration of sentential features. In the next step, our model learns to make coreference decisions for pairs of event mentions based on the previously generated representations. This approach does not

rely on external semantic features, but rather employs a combination of local and sentential features to describe individual event mentions, and combines these intermediate event representations with standard pairwise features for the coreference decision. The model achieves state-of-the-art performance in our experiments on two datasets, one of which is publicly available. Furthermore, we present an analysis of the system errors to identify directions for further research.

## 2 Problem definition

We follow the notion of events from the ACE 2005 dataset (LDC, 2005; Walker et al., 2006). Consider the following example:

> *British bank Barclays had agreed to **buy** Spanish rival Banco Zaragozano for 1.14 billion euros. The **combination** of the banking operations of Barclays Spain and Zaragozano will bring together two complementary businesses and will happen this year, in contrast to Barclays' postponed **merger** with Lloyds.*[1]

Processing these sentences in a prototypical, ACE-style information extraction (IE) pipeline would involve (a) the recognition of entity mentions. In the example, mentions of entities are underlined. Next, (b) words in the text are processed as to whether they elicit an event reference, i.e., event *triggers* are identified and their semantic type is classified. The above sentences contain three event mentions with type *Business.Merge-Org*, shown in boldface. The task of event extraction further requires that (c) participants of recognized events are determined among the entity mentions in the same sentence, i.e., an event's *arguments* are identified and their semantic role wrt. to the event is classified. The three recognized event mentions are:

E1: ***buy***(*British bank Barclays*, *Spanish rival Banco Zaragozano*, *1.14 billion euros*)
E2: ***combination***(*Barclays Spain*, *Zaragozano*, *this year*)
E3: ***merger***(*Barclays*, *Lloyds*)

Often, an IE system involves (d) a disambiguation step of the entity mentions against one another in the same document. This allows to identify that the three mentions of "*Barclays*" in the text as referring to the same real-world entity. The analogous task on the level of event mentions is called (e) event linking (or: event coreference resolution) and is the focus of this paper. Specifically, the task is

---

[1]Based on an example in (Araki and Mitamura, 2015).

to determine that E3 is a singleton reference in this example, while E1 and E2 are coreferential, with the consequence that a document-level event instance can be produced from E1 and E2, listing four arguments (two companies, buying price, and acquisition date).

## 3 Model design

This section first motivates the design decisions of our model for event linking, before going into details about its two-step architecture.

**Event features from literature** So far, a wide range of features has been used for the representation of events and relations for extraction (Zhou et al., 2005; Mintz et al., 2009; Sun et al., 2011) and coreference resolution (Bejan and Harabagiu, 2010; Lee et al., 2012; Liu et al., 2014; Araki and Mitamura, 2015; Cybulska and Vossen, 2015) purposes. The following is an attempt to list the most common classes among them, along with examples:

- **lexical:** surface string, lemma, word embeddings, context around trigger
- **syntactic:** depth of trigger in parse tree, dependency arcs from/to trigger
- **discourse:** distance between coreference candidates, absolute position in document
- **semantic (intrinsic):** comparison of event arguments (entity fillers, present roles), event type of coreference candidates
- **semantic (external):** coreference-candidates similarity in lexico-semantic resources (WordNet, FrameNet) and other datasets (VerbOcean corpus), enrichment of arguments with alternative names from external sources (DBpedia, Geonames)

While lexical, discourse, and intrinsic-semantic features are available in virtually all application scenarios of event extraction/linking, and even syntactic parsing is no longer considered an expensive feature source, semantic features from external knowledge sources pose a significant burden on the application of event processing systems, as these sources are created at high cost and come with limited domain coverage.

Fortunately, recent work has explored the use of a new feature class, *sentential features*, for tackling relation-/event-extraction related tasks with neural networks (Zeng et al., 2014; Nguyen and Grishman, 2015; Chen et al., 2015; dos Santos et al., 2015; Zeng et al., 2015). These approaches have shown that processing sentences with neural models yields representations suitable for IE, which motivates their use in our approach.

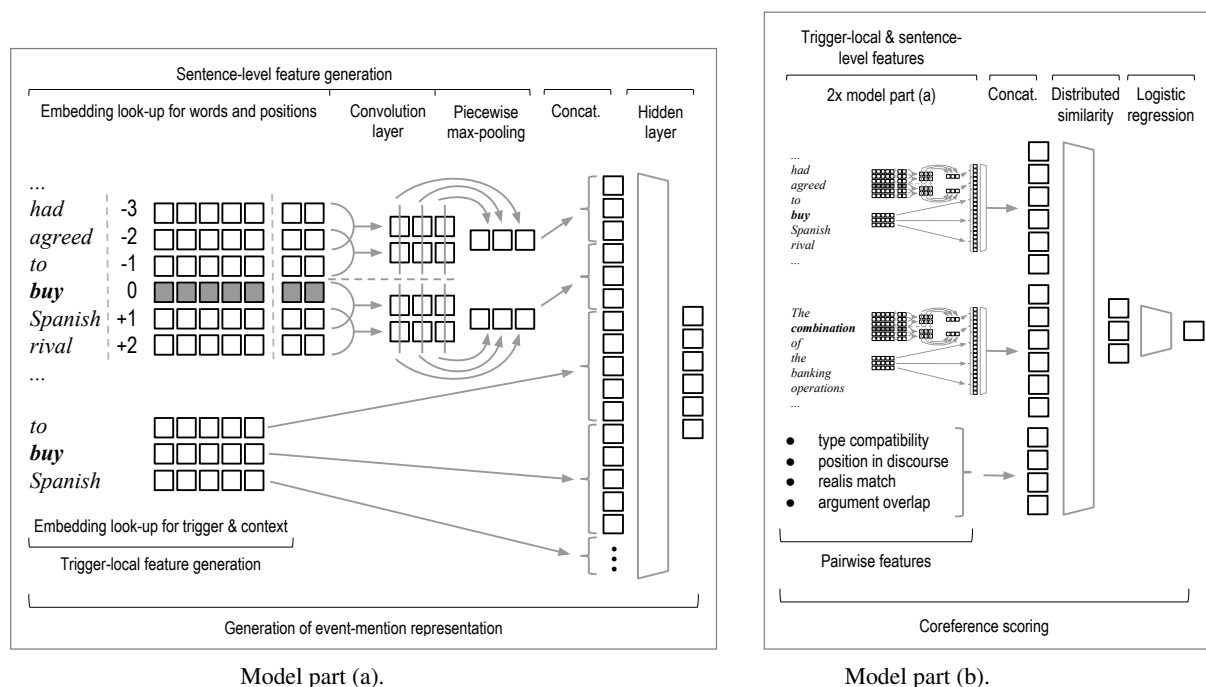

Model part (a).                    Model part (b).

Figure 1: The two parts of the model. The first part computes a representation for a single event mention. The second part is fed with two such event-mention representations plus a number of pairwise features for the input event-mention pair, and calculates a coreference score.

**Data properties** A preliminary analysis of one dataset used in our experiments ($\text{ACE}^{++}$; see Section 5) further motivates the design of our model. We found that 50.97 % of coreferential event-mentions pairs share no arguments, either by mentioning distinct argument roles or because one/both mentions have no annotated arguments. Furthermore, 47.29 % of positive event-mention pairs have different trigger words. It is thus important to not solely rely on intrinsic event properties in order to model event mentions, but to additionally take the surrounding sentence's semantics into account. Another observation regards the distance of coreferential event mentions in a document. 55.42% are more than five sentences apart. This indicates that a locality-based heuristic would not perform well and also encourages the use of sentential features for making coreference decisions.

### 3.1 Learning event representations

The architecture of the model (Figure 1) is split into two parts. The first one aims at adequately representing individual event mentions. As is common in literature, words of the whole sentence of an input event mention are represented as real-valued vectors $v_w^i$ of a fixed size $d_w$, with $i$ being a word's position in the sentence. These *word embeddings*

are updated during model training and are stored in a matrix $W_w \in \mathbb{R}^{d_w \times |V|}$; $|V|$ being the vocabulary size of the dataset.

Furthermore, we take the relative position of tokens with respect to the mention into account, as suggested by (Collobert et al., 2011; Zeng et al., 2014). The rationale is that while the absolute position of learned features in a sentence might not be relevant for an event-related decision, the position of them wrt. the event mention is. Embeddings $v_p^{(\cdot)}$ of size $d_p$ for relative positions are generated in a way similar to word embeddings. Embeddings for words and positions are concatenated into vectors $v_t^{(\cdot)}$ of size $d_t = d_w + d_p$. A sentence with $s$ words is thus represented by a matrix of dimensions $s \times d_t$. This matrix serves as input to a convolution layer.

In order to compress the semantics of $s$ words into a sentence-level feature vector with constant size, the convolution layer applies $d_c$ filters to each window of $n$ consecutive words, calculating $d_c$ features for each $n$-gram of a sentence. For a single filter and particular window, this operation is defined as

$$v_c^i = relu(w_c \cdot v_t^{i:i+n-1} + b_c), \qquad (1)$$

where $w_c \in \mathbb{R}^{n*d_t}$ is a filter, $v_t^{i:i+n-1}$ is the flattened concatenation of vectors $v_t^{(\cdot)}$ for words at

positions $i$ through $i+n-1$, $b_c$ is a bias, and $relu$ is the activation function of a rectified linear unit. In Figure 1, $d_c = 3$ and $n = 2$.

In order to identify the most indicative features in the sentence and to introduce invariance for the absolute position of these, we feed the $n$-gram representations to a max-pooling layer, which identifies the maximum value for each filter. We treat $n$-grams on each side of the trigger word separately, which allows the model to handle multiple event mentions per sentence, similar in spirit to (Chen et al., 2015; Zeng et al., 2015). The pooling is defined as

$$v_m^{j,k} = max(v_c^i), \qquad (2)$$

where $1 \le j \le d_c$ designates a feature, $k \in \{left, right\}$ corresponds to a sentence part, and $i$ runs through the convolution windows of $k$. The output of this step are sentential features $v_{sent} \in \mathbb{R}^{2*d_c}$ of the input event mention.

Additionally, we provide the network with trigger-local, lexical-level features by concatenating $v_{sent}$ with the word embeddings $v_w^{(\cdot)}$ of the trigger word and its left and right neighbor, resulting in $v_{sent+lex} \in \mathbb{R}^{2*d_c+3*d_w}$. This encourages the model to take the lexical semantics of the trigger into account, as these can be a strong indicator for coreference. The result is processed by an additional hidden layer, generating the final event-mention representation $v_e$ with size $d_e$ used for the subsequent event-linking decision:

$$v_e = tanh(W_e v_{sent+lex} + b_e). \qquad (3)$$

### 3.2 Learning coreference decisions

The second part of the model (Figure 1 (b)) processes the representations for two event mentions, and augments these with pairwise comparison features to determine the compatibility of the event mentions. The following features are used, in parentheses we give the feature value for the pair E1, E2 from the example in Section 1:

- Coarse-grained and/or fine-grained event type agreement (*yes*, *yes*)
- Antecedent event is in first sentence (yes)
- (Bagged) distance between event mentions in #sentences/#intermediate event mentions (1, 0)
- Agreement in event modality (yes)
- Overlap in arguments (two shared arguments)

The concatenation of these vectors ($v_{sent+lex+pairw}$) is processed by a single-layer neural network which calculates a distributed similarity of size $d_{sim}$ for the two event mentions:

$$v_{sim} = square(W_{sim} v_{sent+lex+pairw} + b_{sim}). \qquad (4)$$

```
1: procedure GENERATEEXAMPLES(M_d):
2:   M_d = (m_1, ..., m_{|M_d|})
3:   P_d ← ∅
4:   for i = 2, ..., |M_d| do
5:     for j = 1, ..., i − 1 do
6:       P_d ← P_d ∪ {(m_i, m_j)}
7:   return P_d
```

Figure 2: Generation of examples $\mathcal{P}_d$ for a document $d$ with a sequence of event mentions $\mathcal{M}_d$.

```
1: procedure GENERATECLUSTERS(P_d, score):
2:   P_d = {(m_i, m_j)}_{i,j}
3:   score : P_d ↦ (0, 1)
4:   C_d ← {(m_i, m_j) ∈ P_d : score(m_i, m_j) > 0.5}
5:   while ∃(m_i, m_k), (m_k, m_j) ∈ C_d : (m_i, m_j) ∉ C_d do
6:     C_d ← C_d ∪ {(m_i, m_j)}
7:   return C_d
```

Figure 3: Generation of event clusters $\mathcal{C}_d$ for a document $d$ based on the coreference scores from the model. $\mathcal{P}_d$ is the set of all event-mention pairs from a document, as implemented in Figure 2.

The use of the square function as the network's non-linearity is backed by the intuition that for measuring similarity, an invariance under polarity changes is desirable. Having $d_{sim}$ similarity dimensions allows the model to learn multiple similarity facets in parallel; in our experiments, this setup outperformed model variants with different activation functions as well as a cosine-similarity based comparison.

To calculate the final output of the model, $v_{sim}$ is fed to a logistic regression classifier, whose output serves as the coreference score:

$$score = \sigma(W_{out} v_{sim} + b_{out}) \qquad (5)$$

We train the model parameters

$$\theta = \{W_w, W_p, \{w_c\}, \{b_c\}, W_e, b_e, W_{sim}, b_{sim}, W_{out}, b_{out}\} \qquad (6)$$

by minimizing the logistic loss over shuffled mini-batches with gradient descent using Adam (Kingma and Ba, 2014).

### 3.3 Example generation and clustering

We investigated two alternatives for the generation of examples from documents with recognized event mentions. Figure 2 shows the strategy we found to perform best, which iterates over the event mentions of a document and pairs each mention (the "anaphors") with all preceding ones (the "antecedent" candidates). This strategy applies to both training and inference time. Soon et al. (2001) propose an alternative strategy, which during training

|  | ACE | ACE$^{++}$ |
|---|---|---|
| # documents | 599 | 1950 |
| # event instances | 3617 | 7520 |
| # event mentions | 4728 | 9956 |

Table 1: Dataset properties.

| $d_w$ | 300 | $\eta$ | $10^{-5}$ |
|---|---|---|---|
| $d_p$ | 8 | $\beta_1$ | 0.2 |
| $d_c$ | 256 | $\beta_2$ | 0.999 |
| $d_e$ | 50 | $\epsilon$ | $10^{-2}$ |
| $d_{\text{sim}}$ | 2 | batch size | 512 |
| $n$ | 3 | epochs | $\leq 2000$ |
| Dropout | no | $\ell2$ reg. | no |

Table 2: Hyperparameter settings.

creates positive examples only for the closest actual antecedent of an anaphoric event mention with intermediate event mentions serving as negative antecedent candidates. In our experiments, this strategy performed worse than the less elaborate algorithm in Figure 2.

The pairwise coreference decisions of our model induce a clustering of a document's event mentions. In order to force the model to output a consistent view on a given document, a strategy for resolving conflicting decisions is needed. We followed the strategy detailed in Figure 3, which builds the transitive closure of all positive links. Additionally, we experimented with Ng and Gardent (2002)'s "BestLink" strategy, which discards all but the highest-scoring antecedent of an anaphoric event mention. Liu et al. (2014) reported that for event linking, BestLink outperforms naive transitive closure, however, in our experiments (Section 5) we come to a different conclusion.

## 4 Experimental setting, model training

We implemented our model using the TensorFlow framework (Abadi et al., 2015, v0.6), and chose the ACE 2005 dataset (Walker et al., 2006, later: ACE) as our main testbed. The annotation of this corpus focuses on the event types *Conflict.Attack*, *Movement.Transport*, and *Life.Die* reporting about terrorist attacks, movement of goods and people, and deaths of people; but also contains many more related event types as well as mentions of business-relevant and judicial events. The corpus consists of merely 599 documents, which is why we create a second dataset that encompasses these documents and additionally contains 1351 more web documents annotated in an analogous fashion. We refer to this second dataset as ACE$^{++}$. Both datasets are split 9:1 into a development (*dev*) and *test* partition; we further split *dev* 9:1 into a training (*train*) and validation (*valid*) partition.[2] Table 1 lists statistics for the datasets.

There are a number of architectural alternatives

in the model as well as hyperparameters to optimize. Besides the size of intermediate representations in the model ($d_w, d_p, d_c, d_e, d_{\text{sim}}$), we experimented with different convolution window sizes $n$, activation functions for the similarity-function layer in model part (b), whether to use the dual pooling and final hidden layer in model part (a), whether to apply regularization with $\ell2$ penalties or Dropout, and parameters to Adam ($\eta, \beta_1, \beta_2, \epsilon$). We started our exploration of this space of possibilities from previously reported hyperparameter values (Zhang and Wallace, 2015; Chen et al., 2015) and followed a combined strategy of random sampling from the hyperparameter space (180 points) and line search. Optimization was done by training on ACE$^{++}_{train}$ and evaluating on ACE$^{++}_{valid}$. The final settings we used for all following experiments are listed in Table 2. $W_w$ is initialized with pre-trained embeddings of (Mikolov et al., 2013)[3], all other model parameters are randomly initialized. Model training is run for 2000 epochs, after which the best model on the respective *valid* partition is selected.

## 5 Evaluation

This section elaborates on the conducted experiments. First, we compare our approach to state-of-art systems on dataset ACE, after which we report experiments on ACE$^{++}$, where we contrast variations of our model to gain insights about the impact of the utilized feature classes. We conclude this section with an error analysis.

### 5.1 Comparison to state-of-the-art on ACE

Table 3 depicts the performance of our model, trained on ACE$_{train}$, on ACE$_{test}$, along with the performance of state-of-the-art systems from the literature. From the wide range of proposed metrics for the evaluation of coreference resolution, we believe

---

[2]The list of documents in ACE$_{valid}$/ACE$_{test}$ is published here: https://git.io/vwEEP.

[3]https://code.google.com/archive/p/word2vec/

| | BLANC | | | B-CUBED | | | MUC | | | Positive links | | |
|---|---|---|---|---|---|---|---|---|---|---|---|---|
| | 4 ∗ (Precision / Recall / F1 score) in % | | | | | | | | | | | |
| This paper | **71.80** | 75.16 | **73.31** | **90.52** | 86.12 | 88.26 | **61.54** | 45.16 | **52.09** | 47.89 | **56.20** | **51.71** |
| (Liu et al., 2014) | 70.88 | 70.01 | 70.43 | 89.90 | **88.86** | **89.38** | 53.42 | **48.75** | 50.98 | **55.86** | 40.52 | 46.97 |
| (Bejan and Harabagiu, 2010) | — | — | — | 83.4 | 84.2 | 83.8 | — | — | — | 43.3 | 47.1 | 45.1 |
| (Sangeetha and Arock, 2012) | — | — | — | — | — | 87.7 | — | — | — | — | — | — |

Table 3: Event-linking performance of our model & competitors on ACE. Best value per metric in bold.

BLANC (Recasens and Hovy, 2011) has the highest validity, as it balances the impact of positive and negative event-mention links in a document. Negative links and consequently singleton event mentions are more common in this dataset (more than 90 % of links are negative). As Recasens and Hovy (2011) point out, the informativeness of metrics like MUC (Vilain et al., 1995), B-CUBED (Bagga and Baldwin, 1998), and the naive positive-link metric suffers from such imbalance. We still add these metrics for completeness, and because BLANC is not available for all systems.

Unfortunately, there are two caveats to this comparison. Firstly, while a 9:1 train/test split is the commonly accepted way of using ACE, the exact documents in the partitions vary from system to system. Secondly, published methods follow different strategies regarding preprocessing components. While all systems in Table 3 use gold-annotated event-mention triggers, Bejan and Harabagiu (2010) and Liu et al. (2014) use a semantic-role labeler and other tools instead of gold-argument information. We argue that using gold-annotated event mentions is reasonable in order to mitigate error propagation along extraction pipeline and make performance values for the task at hand more informative.

We beat Liu et al. (2014)'s system in terms of F1 score on BLANC, MUC, and positive-links, while their system performs better in terms of B-CUBED. Even when taking into account the caveats mentioned above, it seems justified to assess that our model performs in general on-par with their state-of-the-art system. Their approach involves random-forest classification with best-link clustering and propagation of attributes between event mentions, and is grounded on a manifold of external feature sources, i.e., it uses a "rich set of 105 semantic features". Thus, their approach is strongly tied to domains where these semantic features are available and is potentially hard to port to other text kinds. In contrast, our approach does not depend

| | Model | Dataset | BLANC | | |
|---|---|---|---|---|---|
| | | | (P/R/F1 in %) | | |
| 1) | Section 3 | ACE | 71.80 | **75.16** | **73.31** |
| 2) | Sec. 3 + BestLink | ACE | **75.68** | 69.72 | 72.19 |
| 3) | Section 3 | ACE$^{++}$ | 73.22 | **83.21** | **76.90** |
| 4) | Sec. 3 + BestLink | ACE$^{++}$ | **74.24** | 68.86 | 71.09 |

Table 4: Impact of data amount and clustering.

on resources with restricted domain availability.

Bejan and Harabagiu (2010) propose a non-parametric Bayesian model with standard lexical-level features and WordNet-based similarity between event elements. We outperform their system in terms of B-CUBED and positive-links, which indicates that their system tends to over-merge event mentions, i.e., has a bias against singletons. They use a slightly bigger variant of ACE with 46 additional documents in their experiments.

Sangeetha and Arock (2012) hand-craft a similarity metric for event mentions based on the number of shared entities in the respective sentences, lexical terms, synsets in WordNet, which serves as input to a mincut-based cluster identification. Their system performs well in terms of B-cubed F1, however their paper provides few details about the exact experimental setup.

Another approach with results on ACE was presented by Chen et al. (2009), who employ a maximum-entropy classifier with agglomerative clustering and lexical, discourse, and semantic features, e.g., also a WordNet-based similarity measure. However, they report performance using a threshold optimized on the test set, thus we decided to not include the performance here.

### 5.2 Further evaluation on ACE and ACE$^{++}$

We now look at several aspects of the model performance to gain further insights about it's behavior.

**Impact of dataset size and clustering strategy**
Table 4 shows the impact of increasing the amount of training data (ACE → ACE$^{++}$). This increase

| Pw | Loc | Sen | Dataset | BLANC (P/R/F1 in %) | | |
|---|---|---|---|---|---|---|
| 1) ✓ | | | ACE$^{++}$ | 57.45 | 68.16 | 56.69 |
| 2) ✓ | ✓ | | ACE$^{++}$ | 62.24 | 76.23 | 64.12 |
| 3) ✓ | ✓ | ✓ | ACE$^{++}$ | 73.22 | **83.21** | **76.90** |
| 4) ✓ | | ✓ | ACE$^{++}$ | **82.60** | 70.71 | 74.97 |
| 5) | ✓ | ✓ | ACE$^{++}$ | 59.67 | 66.25 | 61.28 |
| 6) | | ✓ | ACE$^{++}$ | 58.38 | 55.85 | 56.70 |

Table 5: Impact of feature classes; Pw ≡ pairwise features, Loc ≡ trigger-local lexical features, Sen ≡ sentential features.

| Model | Dataset | BLANC (P/R/F1 in %) | | |
|---|---|---|---|---|
| Section 3 | ACE$^{++}$ | **73.22** | 83.21 | **76.90** |
| All singletons | ACE$^{++}$ | 45.29 | 50.00 | 47.53 |
| One instance | ACE$^{++}$ | 4.71 | 50.00 | 8.60 |
| Same type | ACE$^{++}$ | 62.73 | **84.75** | 61.35 |

Table 6: Event-linking performance of our model against naive baselines.

(rows 1, 3) leads to a boost in recall, from 75.16% to 83.21%, at the cost of a small decrease in precision. This indicates that the model can generalize much better using this additional training data.

Looking into the use of the alternative clustering strategy BestLink recommended by Liu et al. (2014), we can make the expected observation of a precision improvement (row 1 vs. 2; row 3 vs. 4), due to fewer positive links being used before the transitive-closure clustering takes place. This is however outweighed by a large decline in recall, resulting in a lower F1 score (73.31 → 72.19; 76.90 → 71.09). The better performance of BestLink in Liu et al.'s model suggests that our model already weeds out many low confidence links in the classification step, which makes a downstream filtering unnecessary in terms of precision, and even counter-productive in terms of recall.

**Impact of feature classes**   Table 5 shows our model's performance when particular feature classes are removed from the model (with re-training), with row 3 corresponding to the full model as described in Section 3. Unsurprisingly, classifying examples with just pairwise features (row 1) results in the worst performance, and adding first trigger-local lexical features (row 2), then sentential features (row 3) subsequently raises both precision and recall. Just using pairwise features and sentential ones (row 4), boosts precision,

which is counter-intuitive at first, but may be explained by a different utilization of the sentential-feature part of the model during training. This part is then adapted to focus more on the trigger-word aspect, meaning the sentential features degrade to trigger-local features. While this allows to reach higher precision (recall that Section 3 finds that more than fifty percent of positive examples have trigger-word agreement), it substantially limits the model's ability to learn other coreference-relevant aspects of event-mention pairs, leading to low recall. Further considering rows 5 & 6, we can conclude that all feature classes indeed positively contribute to the overall model performance.

**Baselines**   The result of applying three naive baselines to ACE$^{++}$ is shown in Table 6. The *all singletons/one instance* baselines predict every input link to be negative/positive, respectively. In particular the all-singletons baseline performs well, due to the large fraction of singleton event mentions in the dataset. The third baseline, *same event*, predicts a positive link whenever there is agreement on the event type, namely, it ignores the possibility that there could be multiple event mentions of the same type in a document which do not refer to the same real-world event, e.g., referring to different terrorist attacks. This baseline also performs quite well, in particular in terms of recall, but shows low precision.

**Error analysis**   We manually investigated a sample of 100 false positives and 100 false negatives from ACE$^{++}$ in order to get an understanding of system errors.

It turns out that a significant portion of the false negatives would involve the resolution of a pronoun to a previous event mention, a very hard and yet unsolved problem. Consider the following examples:

- *"It's crazy that we're **bombing** Iraq. **It** sickens me."*
- *"Some of the slogans sought to rebut **war** supporters' arguments that the protests are unpatriotic. [...] Nobody questions whether **this** is right or not.*

In both examples, the event mentions (trigger words in bold font) are gold-annotated as coreferential, but our model failed to recognize this.

Another observation is that for 17 false negatives, we found analogous cases among the sampled false positives where annotators made a different annotation decision. Consider these examples:

- *The 1860 Presidential **Election**. [...] Lincoln **won** a plurality with about 40% of the vote.*

- *She **lost** her seat in the 1997 **election**.*

Each bullet point has two event mentions (in bold font) taken from the same document and referring to the same event type, i.e., *Personnel.Elect*. While in the first example, the annotators identified the mentions as coreferential, the second pair of mentions is not annotated as such. Analogously, 22 out of the 100 analyzed false positives were cases where the misclassification of the system was plausible to a human rater. This exemplifies that this task has many boundary cases were a positive/negative decision is hard to make even for expert annotators, thus putting the overall performance of all models in Table 3 in perspective.

## 6 Related work

We briefly point out other relevant approaches and efforts from the vast amount of literature.

**Event coreference**   In addition to the competitors mentioned in Section 5, approaches for event linking were presented, e.g., by Chen and Ji (2009), who determine link scores with hand-crafted compatibility metrics for event mention pairs and a maximum-entropy model, and feed these to a spectral clustering algorithm. A variation of the event-coreference resolution task extends the scope to cross-document relations. Cybulska and Vossen (2015) approach this task with various classification models and propose to use a type-specific granularity hierarchy for feature values. Lee et al. (2012) further extend the task definition by jointly resolving entity and event coreference, through several iterations of mention-cluster merge operations. Sachan et al. (2015) describe an active-learning based method for the same problem, where they derive a clustering of entities/events by incorporating bits of human judgment as constraints into the objective function. Araki and Mitamura (2015) simultaneously identify event triggers and disambiguate them wrt. one another with a structured-perceptron algorithm.

**Resources**   Besides the ACE 2005 corpus, a number of other datasets with event-coreference annotation have been presented. Hovy et al. (2013) reports on the annotation process of two corpora from the domains of "violent events" and biographic texts; to our knowledge neither of them is publicly available. OntoNotes (Weischedel et al., 2013) comprises different annotation layers including coreference (Pradhan et al., 2012), however intermin-

gles entity and event coreference. A series of releases of the EventCorefBank corpus (Bejan and Harabagiu, 2010; Lee et al., 2012; Cybulska and Vossen, 2014) combine linking of event mentions within and across documents, for which Liu et al. (2014) report a lack of completeness on the within-document aspect. The ProcessBank dataset (Berant et al., 2014) provides texts with event links from the difficult biological domain.

**Other**   A few approaches to the upstream task of event extraction, while not considering within-document event linking, still utilize discourse-level information or even cross-document inference. For example, Liao and Grishman (2010) showed how the output of sentence-based classifiers can be filtered wrt. discourse-level consistency. Yao et al. (2010) resolved coreferences between events from different documents in order to make a global extraction decision, similar to (Ji and Grishman, 2008) and (Li et al., 2011).

In addition to convolutional neural networks, more types of neural architectures lend themselves to the generation of sentential features. Recently many recursive networks and recurrent ones have been proposed for the task of relation classification, with state-of-the-art results (Socher et al., 2012; Hashimoto et al., 2013; Ebrahimi and Dou, 2015; Xu et al., 2015; Li et al., 2015).

## 7 Conclusion

Our proposed model for the task of event linking achieves state-of-the-art results without relying on external feature sources. We have thus shown that low linking performance, coming from a lack of semantic knowledge about a domain, is evitable. In addition, our experiments give further empirical evidence for the usefulness of neural models for generating latent-feature representations for sentences.

As next steps, we plan to test the model on more datasets and task variations, i.e., in a cross-document setting or for joint trigger identification and coreference resolution. Furthermore, the generation of sentential features from other types of neural networks seems promising. Regarding our medium-term research agenda, we would like to investigate if the model can benefit from more fine-grained information about the discourse structure underlying a text. This could guide the model when encountering the problematic case of pronoun resolution, described in the error analysis.

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
