# Peer review of "Event Linking with Sentential Features from Convolutional Neural Networks"

_CoNLL 2016 — decision unknown_

[Official Review · Reviewer 1 · rating 4 · confidence 4]
soundness 4 · originality 3 · clarity 4 · impact 3 · substance 4 · appropriateness 5 · meaningful comparison 5 · replicability 3 · presentation format Poster

This paper models event linking using CNNs. Given event mentions, the authors
generate vector representations based on word embeddings passed through a CNN
and followed by max-pooling. They also concatenate the resulting
representations with several word embeddings around the mention. Together with
certain pairwise features, they produce a vector of similarities using a
single-layer neural network, and compute a coreference score. 
The model is tested on an ACE dataset and an expanded version with performance
comparable to previous feature-rich systems.
The main contribution of the paper, in my opinion, is in developing a neural
approach for entity linking that combines word embeddings with several
linguistic features. It is interesting to find out that just using the word
embeddings is not sufficient for good performance. Fortunately, the linguistic
features used are limited and do not require manually-crafted external
resources.  

Experimental setting
- It appears that gold trigger words are used rather than predicted ones. The
authors make an argument why this is reasonable, although I still would have
liked to see performance with predicted triggers. This is especially
problematic as one of the competitor systems used predicted triggers, so the
comparison isn't fair. 
- The fact that different papers use different train/test splits is worrisome.
I would encourage the authors to stick to previous splits as much as possible. 

Unclear points
- The numbers indicating that cross-sentential information is needed are
convincing. However, the last statement in the second paragraph (lines 65-70)
was not clear to me.
- Embeddings for positions are said to be generaties "in a way similar to word
embeddings". How exactly? Are they randomly initialized? Are they lexicalized?
It is not clear to me why a relative position next to one word should have the
same embedding as a relative position next to a different word.
- How exactly are left vs right neighbors used to create the representation
(lines 307-311)? Does this only affect the max-pooling operation?
- The word embeddings of one word before and one word after the trigger words
are appended to it. This seems a bit arbitrary. Why one word before and after
and not some other choice?  
- It is not clear how the event-mention representation v_e (line 330) is used?
In the following sections only v_{sent+lex} appear to be used, not v_e.
- How are pairwise features used in section 3.2? Most features are binary, so I
assume they are encoded as a binary vector, but what about the distance feature
for example? And, are these kept fixed during training?

Other issues and suggestions
- Can the approach be applied to entity coreference resolution as well? This
would allow comparing with more previous work and popular datasets like
OntoNotes. 
- The use of a square function as nonlinearity is interesting. Is it novel? Do
you think it has applicability in other tasks?
- Datasets: one dataset is publicly available, but results are also presented
with ACE++, which is not. Do you have plans to release it? It would help other
researchers compare new methods. At least, it would have been good to see a
comparison to the feature-rich systems also on this dataset.
- Results: some of the numbers reported in the results are quite close.
Significance testing would help substantiating the comparisons.
- Related work: among the work on (entity) coreference resolution, one might
mention the neural network approach by Wiseman et al. (2015)  

Minor issues
- line 143, "that" is redundant. 
- One of the baselines is referred to as "same type" in table 6, but "same
event" in the text (line 670).        

Refs
- Learning Anaphoricity and Antecedent Ranking Features for Coreference
Resolution. Sam Wiseman, Alexander M. Rush, Jason Weston, and Stuart M.
Shieber. ACL 2015.

[Official Review · Reviewer 2 · rating 3 · confidence 4]
soundness 3 · originality 3 · clarity 5 · impact 2 · substance 4 · appropriateness 5 · meaningful comparison 3 · replicability 3 · presentation format Oral Presentation

This paper presents a model for the task of event entity linking, where they
propose to use sentential features from CNNs in place of external knowledge
sources which earlier methods have used. They train a two-part model: the first
part learns an event mention representation, and the second part learns to
calculate a coreference score given two event entity mentions.

The paper is well-written, well-presented and is easy to follow. I rather like
the analysis done on the ACE corpus regarding the argument sharing between
event coreferences. Furthermore, the analysis on the size impact of the
dataset is a great motivation for creating their ACE++ dataset. However, there
are a few
major issues that need to be addressed:

- The authors fail to motivate and analyze the pros and cons of using CNN for
generating mention representations. It is not discussed why they chose CNN and
there are no comparisons to the other models (e.g., straightforwardly an RNN).
Given that the improvement their model makes according various metrics against
the
state-of-the-art is only 2 or 3 points on F1 score, there needs to be more
evidence that this architecture is indeed superior.

- It is not clear what is novel about the idea of tackling event linking with
sentential features, given that using CNN in this fashion for a classification
task is not new. The authors could explicitly point out and mainly compare to
any existing continuous space methods for event linking. The choice of methods
in Table 3 is not thorough enough.

- There is no information regarding how the ACE++ dataset is collected. A major
issue with the ACE dataset is its limited number of event types, making it too
constrained and biased. It is important to know what event types ACE++ covers.
This can also help support the claim in Section 5.1 that 'other approaches are
strongly tied to the domain where these semantic features are availableâ¦our
approach does not depend on resources with restrictedâ¦', you need to show
that those earlier methods fail on some dataset that you succeed on. Also,
for enabling any meaningful comparison in future, the authors should think
about making this dataset publicly available.

Some minor issues:
- I would have liked to see the performance of your model without gold
references in Table 3 as well.

- It would be nice to explore how this model can or cannot be augmented with a
vanilla coreference resolution system. For the specific example in line 687,
the off-the-shelf CoreNLP system readily links 'It' to 'bombing', which can be
somehow leveraged in an event entity linking baseline.

- Given the relatively small size of the ACE dataset, I think having a
compelling model requires testing on the other available resources as well.
This further motivates working on entity and event coreference simultaneously.
I also believe that testing on EventCorefBank in parallel with ACE is
essential. 

- Table 5 shows that the pairwise features have been quite effective, which
signals that feature engineering may still be crucial for having a competitive
model (at least on the scale of the ACE dataset). One would wonder which
features were the most effective, and why not report how the current set was
chosen and what else was tried.